# PhysDiff-VTON: Cross-Domain Physics Modeling and Trajectory Optimization for Virtual Try-On

**Shibin Mei**
Huawei
shibin.mei1027@gmail.com

**Bingbing Ni** *
Shanghai Jiao Tong University
nibingbing@sjtu.edu.cn

## Abstract

We present PhysDiff-VTON, a diffusion-based framework for image-based virtual try-on that systematically addresses the dual challenges of garment deformation modeling and high-frequency detail preservation. The core innovation lies in integrating physics-inspired mechanisms into the diffusion process: a pose-guided deformable warping module simulates fabric dynamics by predicting spatial offsets conditioned on human pose semantics, while wavelet-enhanced feature decomposition explicitly preserves texture fidelity through frequency-aware attention. Further enhancing generation quality, a novel sampling strategy optimizes the denoising trajectory via least action principles, enforcing temporal coherence, spatial smoothness, and multi-scale structural consistency. Comprehensive evaluations across multiple datasets demonstrate significant improvements in both geometric plausibility and perceptual quality compared to existing approaches. The framework establishes a new paradigm for synthesizing photorealistic try-on images that adhere to physical constraints while maintaining intricate garment details, advancing the practical applicability of diffusion models in fashion technology.

## 1 Introduction

Recent advances in diffusion models [12, 31] have revolutionized image-based virtual try-on (VTON) [19, 40, 23, 8] by enabling high-fidelity synthesis of garment-person interactions [18, 4]. However, existing approaches still struggle to reconcile two critical aspects, that is, preserving high-frequency texture details while accommodating complex fabric deformations, and enforcing physical plausibility in garment dynamics under diverse human poses. Current pipelines often rely on global affine transformations or heuristic warping [44], which inadequately model localized nonlinear deformations caused by pose variations [5, 40]. Meanwhile, the stochastic nature of diffusion sampling introduces unintended artifacts in fine textures, particularly under occlusion or extreme articulation [34].

Recent arts [6, 42, 20] have witnessed concerted efforts to address these challenges. IDM-VTON [4] pioneers dual encoders to decouple garment semantics and structural features, achieving notable improvements in texture fidelity through cross-attention fusion of high-level semantic embeddings and low-level UNet [32] features. However, its linear blending strategy rigidly combines these features without modeling temporal fabric dynamics, causing discontinuous wrinkle transitions in articulated poses. GP-VTON [40] introduces geometric parsing to resolve coarse misalignments via localized part-based warping, yet its component-level deformation neglects microscale texture continuity across seam boundaries, exacerbating pattern discontinuities in plaid or striped fabrics. Parallel wavelet-based approaches leverage frequency-domain decomposition to preserve high-frequency details, but their isolated spectral processing fails to synchronize with pose-dependent deformation fields, resulting in physically inconsistent texture densities under stretching [38]. These innovations,

---

*Corresponding author.

39th Conference on Neural Information Processing Systems (NeurIPS 2025).

while advancing specific aspects, collectively expose a systemic limitation, that is, the absence of unified frameworks that bridge data-driven feature learning with continuum mechanics principles governing fabric deformation. Consequently, existing methods enforce artificial separations between geometric warping and texture synthesis, propagating errors that manifest as either over-smoothed details or biomechanically implausible drapery.

The dual demands of geometrically plausible deformation and texture-accurate synthesis in virtual try-on necessitate a co-design framework that bridges physics-inspired mechanics with spectral fidelity preservation [48]. Traditional warping methods suffer from irreversible high-frequency distortions when handling complex poses, as their rigid geometric transformations conflict with the Nyquist-Shannon sampling theorem governing textile patterns [13]. Our pose-guided deformable warping addresses this by simulating strain-dependent fabric dynamics through pose embeddings. Here, pose embeddings are employed to predict spatially variant offsets that preserve local curvature continuity, while cross-attention between pose maps and garment segments injects material adaptability, where stiffer fabrics yield smaller offsets for identical movements, emulating real-world drape physics [22]. Crucially, this physics-aware deformation creates a geometrically stable foundation for subsequent wavelet-enhanced texture preservation. Conventional UNet architectures inherently attenuate high-frequency signals through successive downsampling, exacerbating texture erosion during iterative denoising. To counteract this, we integrate Haar wavelet transforms into skip connections, explicitly decoupling high-frequency components, such as edge gradients and micro-textures, from low-frequency shape approximations. A frequency-gated attention mechanism dynamically recalibrates subband contributions, that is, amplifying directional harmonics along deformation axes while suppressing orthogonally misaligned noise [38]. This dual-domain synergy ensures that geometric transformations adhere to continuum mechanics while spectral constraints enforce texture Nyquist compliance—a fundamental advance over isolated spatial or frequency-based approaches.

Building upon the geometrically consistent deformations and spectrally preserved textures achieved by our cross-domain framework, the final pillar, potential-regularized path optimization (PRPO), addresses the temporal and structural coherence challenges inherent in iterative diffusion sampling. While pose-guided warping ensures fabric dynamics obey Newtonian principles and wavelet decomposition maintains Nyquist-compliant textures, the stochastic denoising trajectory may still accumulate errors across timesteps, manifesting as anatomically implausible wrinkles or discontinuous fabric flows during arm articulation. PRPO reinterprets this sampling process through the lens of continuum mechanics [26], formulating it as a variational problem that minimizes an action functional encompassing three physics-inspired potentials, that is, temporal incoherence penalty via inter-timestep smoothness, spatial irregularities constraint via total variation, and structural inconsistencies [45] suppression via multiscale self-similarity. By deriving gradient corrections from these potentials, PRPO steers the diffusion path toward energetically favorable states. This completes our physics-integrated control loop, where the first two modules establish geometric-spectral foundations, while PRPO orchestrates their synergistic evolution across the generative trajectory, achieving texture fidelity even under extreme articulation.

Our PhysDiff-VITON framework pioneers physics-integrated diffusion for virtual try-on, establishing new standards in geometric fidelity and spectral authenticity. Our contributions can be summarized as,

- We develop PhysDiff-VITON, a framework that unifies continuum mechanics principles with diffusion dynamics through a pose-conditioned deformation module and wavelet-constrained texture synthesis, enabling simulation of fabric dynamics under complex articulations.

- We propose adaptive deformation preserving curvature continuity, wavelet-constrained synthesis maintaining texture fidelity, and trajectory regularization through energy-minimized sampling.

- Comprehensive experiments confirm the superiority of our method.

## 2 Cross-Domain Physics Modeling

In this section, we present the PhysDiff-VTON framework, which systematically integrates physics-inspired deformation modeling in Sec. 2.2, spectral fidelity preservation in Sec. 2.3. We start with a preliminary in Sec. 2.1 that formalizes the diffusion process for virtual try-on.

## 2.1 Preliminary: Diffusion Models for Virtual Try-On

Diffusion models formulate image synthesis as an iterative denoising process governed by stochastic differential equations [12]. For virtual try-on, given a source person image $\mathbf{I}_p \in \mathbb{R}^{H \times W \times 3}$ and a garment image $\mathbf{I}_g \in \mathbb{R}^{H \times W \times 3}$, the goal is to generate $\hat{\mathbf{I}}$ where $\mathbf{I}_g$ is realistically worn by the person in $\mathbf{I}_p$ while preserving pose and texture details. The forward diffusion process progressively corrupts the target image $\mathbf{x}_0$ with Gaussian noise across $T$ timesteps,

$$q(\mathbf{x}_t|\mathbf{x}_{t-1}) = \mathcal{N}\left(\mathbf{x}_t; \sqrt{1-\beta_t}\mathbf{x}_{t-1}, \beta_t\mathbf{I}\right), \quad q(x_{1:T}|x_0) = \prod_{t=1}^{T} q(x_t|x_{t-1}) \tag{1}$$

where $\beta_t$ controls the noise schedule. The reverse process learns to iteratively denoise $\mathbf{x}_T \sim \mathcal{N}(0, \mathbf{I})$ by estimating the score function $\epsilon_\theta(\mathbf{x}_t, t, C)$ conditioned on inputs $C = \{\mathbf{I}_p, \mathbf{I}_g, c_o\}$, where $c_o$ represents other extra conditions, such as text, mask, and pose keypoints. The conditional generation objective can be denoted as,

$$\mathcal{L} = \mathbb{E}_{t,\mathbf{x}_0,\epsilon}\left[\|\epsilon_\theta(\mathbf{x}_t, t, C) - \epsilon\|_2^2\right] \tag{2}$$

The reverse sampling process is to gradually restore the data distribution through the trained denoising network $\epsilon_\theta$.

Unlike generic image synthesis, virtual try-on requires solving a compositional inpainting problem, where replaces the original garment region in $\mathbf{I}_p$ with $\mathbf{I}_g$ while maintaining photometric consistency in non-target regions, biomechanical plausibility in garment deformation, and high-frequency texture fidelity under perspective distortion. This necessitates specialized conditioning mechanisms. Typical implementations [18, 4] encode $\mathbf{I}_g$ through a UNet-based garment encoder $\mathcal{E}_g$ to produce multi-scale features $\{\mathbf{f}_l^g\}_{l=1}^L$, which are fused with person features $\mathbf{f}^p$ via cross-attention layers and self-attention layers in the diffusion model.

The framework of our virtual try-on model is consistent with IDM-VITON [4], which is based on diffusion models, and applies two separate modules to extract semantic information from garment images and encode it into the base UNet. We utilizes a visual encoder to extract high-level semantic information from garment images and a parallel UNet, i.e., GarmentNet, to extract low-level features, thus preserving details. In addition, detailed text prompts are provided for both garment and person images to enhance the authenticity of the generated images.

## 2.2 Pose-Guided Deformable Fabric Dynamics Modeling

The geometric discrepancy between canonical garment representations and dynamically posed human bodies introduces two fundamental challenges in virtual try-on that is, irreversible texture distortion caused by rigid spatial transformations, and physically implausible deformation due to neglecting material-dependent strain-stress relationships. Traditional approaches relying on affine transformations or thin-plate splines (TPS) [10, 40] impose global smoothness constraints incompatible with localized fabric dynamics, while convolutional warping lacks explicit mechanisms to encode human kinematics. Our physics-aware deformation field addresses these limitations through pose-semantic conditioned offset prediction that emulates strain-dependent displacement.

We first construct a pose embedding $\mathbf{E}_p \in \mathbb{R}^{D_p \times H \times W}$ through cascaded residual blocks from the pose image. We will integrate the pose information into the garment features of different layers. Given garment features $\mathbf{F}_g \in \mathbb{R}^{C \times H \times W}$ extracted via pre-trained UNet in a certain layer, cross-modal attention computes pixel-wise affinity between human pose and fabric regions,

$$\alpha_{ij} = \text{Softmax}\left(\frac{\mathbf{W}_q\mathbf{E}_p^{(i)} \cdot (\mathbf{W}_k\mathbf{F}_g^{(j)})^\top}{\sqrt{D}}\right) \tag{3}$$

where $\mathbf{W}_q, \mathbf{W}_k$ project features to query-key space. The attention map $\mathbf{A} \in \mathbb{R}^{H \times W \times K}$ aggregates pose-specific deformation cues, guiding the prediction of strain-aware offsets,

$$\Delta p = \mathcal{G}(\mathbf{F}_g \oplus (\mathbf{A} \odot \mathbf{E}_p)), \tag{4}$$

Here $\odot$ denotes element-wise multiplication that emphasizes pose-relevant features, and $\oplus$ is a simple concatenation. The offset predictor $\mathcal{G}$ learns material-dependent deformation patterns, where stiffer fabrics exhibit smaller $\|\Delta p\|$ for equivalent joint movements, as observed in real draping physics.

The warped garment features $\tilde{\mathbf{F}}_g$ are computed via deformable convolution:

$$\tilde{\mathbf{F}}_g(x) = \sum_{k=1}^{K} w_k \cdot \mathbf{F}_g(x + p_k + \Delta p_k) \tag{5}$$

where $p_k$ enumerates $K$ sampling locations in the regular grid, and $w_k$ denotes adaptive weights.

Pose-conditioned offset prediction fundamentally differs from prior geometric warping [37] in that it integrates human pose into deformation mechanics. Multi-layer attention enables hierarchical modeling, where global pose changes (e.g., arm elevation) are captured in low-resolution layers, while high-resolution branches handle local wrinkles. Compared to occlusion-agnostic warping in [49], our approach implicitly handles self-occlusions through strain-dependent displacement, where occluded regions automatically receive smaller $\|\Delta p\|$ due to attenuated attention responses. Compared to TPS-based methods [10], our data-driven approach better handles non-linear wrinkles when trained on diverse poses. The network automatically learns to amplify offsets near bending joints while suppressing unrealistic stretching in rigid areas.

## 2.3 Wavelet-Enhanced Spectral Fidelity Preservation

High-frequency texture erosion poses a fundamental challenge in diffusion-based virtual try-on, as iterative downsampling operations in UNet architectures progressively attenuate directional gradients and micro-patterns (e.g., plaid alignments or embroidery stitches) [17, 33]. To address this, we propose a Haar wavelet transform module embedded in skip connections, which explicitly decouples and reinforces high-frequency components throughout the denoising trajectory. The key insight stems from the observation that conventional spatial attention mechanisms exhibit limited discriminative capacity in preserving Nyquist-critical frequencies [15], those carrying essential perceptual information about textile microstructures.

Our implementation begins with a fixed Haar wavelet decomposition applied to intermediate feature maps $\mathbf{X} \in \mathbb{R}^{B \times C \times H \times W}$ after each downsampling layer. The transform applies separable 1D convolution along row and column dimensions using low-pass ($\mathbf{W}_L = [1, 1]$) and high-pass ($\mathbf{W}_H = [1, -1]$) filters. This yields four subbands,

$$\{\mathbf{LL}, \mathbf{LH}, \mathbf{HL}, \mathbf{HH}\} = \mathrm{DWT}(\mathbf{X}), \tag{6}$$

where $\mathbf{LL}$ captures low-frequency approximations, while $\{\mathbf{LH}, \mathbf{HL}, \mathbf{HH}\}$ encode horizontal, vertical, and diagonal high-frequency details, respectively. These subbands are concatenated and processed by a frequency-gated attention mechanism,

$$\mathbf{A} = \sigma(\mathbf{Conv}_1(\mathrm{GN}(\mathbf{Conv}_2(\mathrm{Concat}(\mathbf{LL}, \mathbf{LH}, \mathbf{HL}, \mathbf{HH}))))), \tag{7}$$

where $\sigma$ denotes the sigmoid function, GN represents group normalization, and $\mathbf{Conv}_i$ are convolutional layers. The attention mask $\mathbf{A} \in [0, 1]^{B \times C \times H \times W}$ dynamically amplifies critical frequency components based on local texture complexity. The final enhanced features are computed through residual modulation,

$$\tilde{\mathbf{X}} = \mathbf{X} + \mathbf{A} \odot \mathbf{X}, \tag{8}$$

where $\odot$ denotes element-wise multiplication. This formulation ensures gradient stability while enabling explicit high-frequency preservation. The Haar basis proves particularly effective due to its compact support and directional sensitivity. These properties align with the anisotropic nature of garment textures under deformation.

The essence of this approach lies in transforming spatial-domain image information into the frequency domain through wavelet basis functions such as Haar or Daubechies [46], decomposing garment images into low-frequency components (representing global contours and color distributions) and high-frequency components (capturing edge details and micro-textures). For instance, in a plaid shirt scenario, low-frequency components preserve the general grid arrangement pattern, while high-frequency components precisely record the sharpness of each grid edge and microscopic structures at intersections. Introducing wavelet transforms into the model equips the network with a "computational microscope", compelling continuous attention to these vulnerable details during generation. By designing frequency-aware attention mechanisms in the wavelet domain, the model dynamically amplifies influential frequency bands, enhancing weights on high-frequency subbands when processing glossy silk textures, while suppressing them in smooth regions of plain T-shirts to

prevent noise introduction. This adaptive mechanism ensures optimal detail fidelity across diverse garment categories.

**Synergy between Pose-Guided Deformation and Wavelet-Enhancement**

The synergistic interaction between components further amplifies their respective advantages. The explicit preservation of high-frequency details through wavelet transforms enriches local texture information for deformable convolution, maintaining microstructural continuity during deformation—for example, preventing line fractures in plaid patterns when simulating skirt sway. Conversely, pose-guided warping guarantees proper spatial distribution of high-frequency textures. When garments stretch due to pose variations (e.g., T-shirt print widening during arm extension), high-frequency texture densities adapt accordingly [14]. This synchronized coordination between detail preservation and deformation constitutes the cornerstone of photorealistic generation.

# 3 PRPO for Trajectory Optimization

In this section, we elaborate the trajectory optimization into a unified diffusion paradigm in Sec. 3.1 and potential function design for our newly proposed sampling method in Sec. 3.2.

## 3.1 Potential-Regularized Path Optimization

The stochastic denoising trajectory of diffusion models, while effective in exploring the data manifold, may introduce path oscillations that violate physical priors inherent to virtual try-on tasks. For instance, abrupt changes in latent states across timesteps can lead to discontinuous fabric flows or implausible wrinkles during arm articulation. Inspired by the *principle of least action* [36] in physics, where dynamic systems evolve along paths minimizing an action functional [25, 41]. We reinterpret the diffusion sampling process as a variational optimization problem. This perspective allows for injecting physics-inspired constraints into the generative trajectory, steering it toward states that balance data fidelity with physical plausibility.

**Action Functional.** Define the action functional $\mathcal{S}[x(t)]$ over the diffusion path $x(t)$ from noise $x(T)$ to clean data $x(0)$:

$$\mathcal{S}[x(t)] = \underbrace{\int_0^T \|s_\theta(x(t), t) - \nabla_x \log q_t(x(t))\|^2 dt}_{\text{Dynamic Matching}} + \underbrace{\lambda \int_0^T E(x(t)) dt}_{\text{Potential Regularization}} + \underbrace{\sigma \int_0^T \|\xi(t)\|^2 dt,}_{\text{Stochastic Control}} \quad (9)$$

where $s_\theta$ is the learned score function, $E(x)$ denotes the potential energy encoding physical constraints, and $\xi(t)$ controls stochasticity. The first term enforces consistency with the learned data manifold, the second imposes task-specific physical priors, and the third regulates exploration-exploitation trade-offs.

**Variational Optimization.** Applying variational calculus to minimize $\mathcal{S}$ yields the modified reverse-time SDE:

$$dx = \underbrace{\left[ f(x, t) - g(t)^2 s_\theta(x, t) \right] dt}_{\text{Standard Reverse SDE}} + \underbrace{\lambda g(t)^2 \nabla_x E(x) dt}_{\text{Potential Gradient}} + \underbrace{\sigma g(t) d\bar{w}}_{\text{Controlled Noise}}, \quad (10)$$

where $f(x, t)$ and $g(t)$ are drift and diffusion coefficients from the forward process. The potential gradient term $\lambda g(t)^2 \nabla_x E(x)$ explicitly corrects the trajectory toward low-energy states, while $\sigma g(t) d\bar{w}$ injects annealed noise to avoid local minima.

**Discretized Sampling.** Integrating (10) via the Euler-Maruyama scheme gives the PRPO update rule:

$$x_{t-1} = \underbrace{\frac{1}{\sqrt{\alpha_t}} \left( x_t - \frac{\beta_t}{\sqrt{1 - \bar{\alpha}_t}} \epsilon_\theta(x_t, t) \right)}_{\text{Deterministic Update}} + \underbrace{\lambda \beta_t \nabla_x E(x_t)}_{\text{Potential Correction}} + \underbrace{\sigma \sqrt{\beta_t} z}_{\text{Annealed Noise}}, \quad (11)$$

where $\alpha_t$, $\beta_t$ are DDPM scheduling parameters, and $z \sim \mathcal{N}(0, I)$. The noise scale $\sigma$ decays as $\sigma(t) = \sigma_{\max} \exp(-k(T - t))$ to prioritize exploration early and refinement late.

The potential gradient $\nabla E$ introduces *second-order guidance* beyond the score function's first-order manifold approximation, suppressing non-physical oscillations (e.g., jagged edges in plaid patterns).

---

**Algorithm 1** Potential-Regularized Diffusion Sampling (PRPO)

---

**Require:** Pretrained score model $\epsilon_\theta$, potential function $E(x)$, initial noise $x_T \sim \mathcal{N}(0, I)$

 1: **for** $t = T$ **to** 1 **do**
 2:     Compute $\alpha_t = \prod_{s=1}^{t}(1 - \beta_s)$, $\beta_t = 1 - \alpha_t/\alpha_{t-1}$
 3:     Predict noise $\epsilon_t = \epsilon_\theta(x_t, t)$
 4:     Deterministic update: $\mu_t = \frac{1}{\sqrt{\alpha_t}}\left(x_t - \frac{\beta_t}{\sqrt{1-\bar{\alpha}_t}}\epsilon_t\right)$
 5:     Compute potential gradient: $\nabla E = \nabla_x E(x_t)$
 6:     Apply correction: $\mu_t \leftarrow \mu_t + \lambda\beta_t\nabla E$
 7:     Sample noise scale: $\sigma_t = \sigma_{\max}\exp(-k(T-t))$
 8:     Inject noise: $x_{t-1} = \mu_t + \sigma_t\sqrt{\beta_t}z$ where $z \sim \mathcal{N}(0, I)$
 9: **end for**
10: **return** Denoised sample $x_0$ =0

---

The annealing noise schedule preserves diversity while ensuring final sample coherence, which is crucial for resolving ambiguous cases like occluded garment regions. Alg. 1 details the PRPO sampling process. Lines 2-4 implement the standard DDPM prediction, while Lines 5-6 compute the potential gradient correction. The adaptive noise injection in Lines 7-8 ensures progressive transition from stochastic exploration to deterministic refinement. Notably, PRPO maintains compatibility with existing diffusion frameworks by simply augmenting the sampling step with physics-aware corrections.

### 3.2 Potential Function Design

The efficacy of PRPO critically depends on the design of potential energy functions $E(x)$ that encode domain-specific physical priors. We derive three complementary potentials addressing temporal coherence, spatial regularity, and structural consistency based on the fundamental requirements for photorealistic virtual try-on. Each potential component is derived from first principles of stochastic dynamics and image statistics, ensuring their complementary roles in virtual try-on generation.

**Inter-Timestep Smoothness Potential** Let $z_t \in \mathbb{R}^d$ denote the latent state at timestep $t$ in the diffusion process, with the forward Markov chain defined by:

$$q(z_t|z_{t-1}) = \mathcal{N}(z_t; \sqrt{\alpha_t}z_{t-1}, (1 - \alpha_t)I)$$

The reverse process approximates the true posterior $q(z_{t-1}|z_t)$ through variational inference. To suppress abrupt transitions between timesteps – which manifest as discontinuous fabric flows – we impose the smoothness potential:

$$E_{\text{smooth}}(z_t) = \lambda_t\|z_t - \mathbb{E}[z_{t+1}|z_t]\|_2^2$$

where $\mathbb{E}[z_{t+1}|z_t] = \sqrt{\alpha_{t+1}}z_t$ follows the forward process expectation, and $\lambda_t = \lambda_0 e^{-\gamma t}$ implements time-decaying regularization strength. This term minimizes deviations from the theoretical diffusion trajectory, effectively damping high-frequency oscillations in the denoising path [20]. For articulated garments, such temporal coherence ensures wrinkle formation adheres to progressive drapery dynamics rather than erratic noise artifacts.

**Total Variation Spatial Potential** Natural images exhibit *heavy-tailed gradient distributions* – predominantly smooth regions punctuated by sparse edges. To replicate these statistics in generated garments, we define:

$$E_{\text{TV}}(z_t) = \sum_{i,j}(|\nabla_h z_t[i,j]| + |\nabla_v z_t[i,j]|)$$

where $\nabla_h$ and $\nabla_v$ denote horizontal/vertical finite differences. This total variation (TV) term imposes a piecewise smoothness prior $p(z_t) \propto e^{-\lambda E_{\text{TV}}(z_t)}$, steering solutions toward texture-continuous regions separated by sharp edges [35]. In virtual try-on, $E_{\text{TV}}$ proves critical for preserving high-frequency details like embroidery patterns and fabric seams while suppressing salt-and-pepper noise in homogeneous areas.

**Multiscale Self-Similarity Potential** Leveraging the inherent hierarchy in diffusion processes, we enforce structural consistency across scales:

$$E_{\text{MS}}(z_t) = \sum_{s\in\mathcal{S}}\|\mathcal{D}_s(z_t) - \mathbb{E}[\mathcal{D}_s(z_{t-1})|z_t]\|^1$$

Table 1: Quantitative comparisons on VITON-HD and DressCode test sets. PhysDiff-VTON demonstrates superior performance in both low-level similarity and high-level semantic similarity (LPIPS, SSIM, CLIP-I) and image fidelity (FID). Several GAN-based virtual try-on methods and Diffusion-based virtual try-on methods are introduced to compare with our proposed PhysDiff-VTON. **Bold** denotes the best score for each metric.

| Dataset | VITON-HD | | | | DressCode | | | |
|---|---|---|---|---|---|---|---|---|
| Method | LPIPS $\downarrow$ | SSIM $\uparrow$ | FID $\downarrow$ | CLIP-I $\uparrow$ | LPIPS $\downarrow$ | SSIM $\uparrow$ | FID $\downarrow$ | CLIP-I $\uparrow$ |
| GAN-based methods | | | | | | | | |
| HR-VITON | 0.115 | 0.883 | 9.70 | 0.832 | 0.112 | 0.910 | 21.42 | 0.771 |
| GP-VTON | 0.105 | **0.898** | 6.43 | 0.874 | 0.484 | 0.780 | 55.21 | 0.628 |
| Diffusion-based methods | | | | | | | | |
| LaDI-VTON | 0.156 | 0.872 | 8.85 | 0.834 | 0.149 | 0.905 | 16.54 | 0.803 |
| DCI-VTON | 0.166 | 0.856 | 8.73 | 0.840 | 0.162 | 0.893 | 17.63 | 0.777 |
| StableVITON | 0.133 | 0.885 | 6.52 | 0.871 | 0.107 | 0.910 | 14.37 | 0.866 |
| IDM-VITON | 0.102 | 0.870 | 6.29 | 0.883 | 0.062 | 0.920 | 8.64 | 0.904 |
| PhysDiff(Ours) | **0.093** | 0.881 | **6.21** | **0.894** | **0.055** | **0.932** | **8.27** | **0.918** |

Here, $\mathcal{D}_s$ denotes downsampling by factor $s$, and $\mathbb{E}[\mathcal{D}_s(z_{t-1})|z_t] = \sqrt{\alpha_{t-1}}\mathcal{D}_s(z_t)$ propagates coarse-scale expectations. This potential ensures localized details (e.g., sleeve pleats) remain geometrically consistent with global garment structure [27]. For complex poses, it prevents anatomically implausible distortions by maintaining cross-scale correspondences in deformation fields.

**Unified Variational Perspective** The composite potential $E(x) = E_{\text{smooth}} + E_{\text{TV}} + E_{\text{MS}}$ rectifies the reverse process distribution:

$$p_\theta(z_{0:T}) \propto \prod_{t=1}^{T} p_\theta(z_{t-1}|z_t) \cdot \prod_t e^{-\lambda E(z_t)}$$

This Bayesian formulation injects physics-aware priors into the generative trajectory without altering the base diffusion model. The temporal term $E_{\text{smooth}}$ governs fabric dynamics continuity, $E_{\text{TV}}$ enforces Nyquist-compliant textures, and $E_{\text{MS}}$ maintains anthropometric plausibility across scales, which collectively addressing the trilemma of garment realism.

### 3.3 Implementation Details

We employ the Adam optimizer with a fixed learning rate of $1 \times 10^{-5}$ for 130 training epochs, requiring approximately 95 hours on 4×H800 GPUs. Our data augmentation strategy aligns with Stable-VITON [18], featuring a 0.5 probability of horizontal flipping and 0.5 probability of random affine transformations. During inference, we utilize the PRPO sampler with 30 denoising steps and maximum strength ($\eta = 1.0$), initiating from random noise while disregarding masked regions in the input person image. For classifier-free guidance, inspired by IDM-VITON [4] and SpaText [2], we jointly condition the model using low-level garment features and high-level semantic features from IP-Adapter [43]. Distinctively, we implement pose-guided feature warping through a learnable deformation module $\mathcal{D}(\cdot)$ that modulates garment features $\mathbf{F}_g$ based on pose map $\mathbf{P}$: $\tilde{\mathbf{F}}_g = \mathcal{D}(\mathbf{F}_g|\mathbf{P})$. The guidance scale $w$ is set to 2.0. The garment features are extracted from the first 10 channels of a pretrained diffusion bottleneck layer of UNet, which are subsequently injected into the target diffusion model after pose-aware deformation. To enhance detail preservation, we integrate wavelet-transform-based frequency selection modules after each cross-attention layer in the UNet downsampling blocks, operating in the Haar wavelet domain to perform frequency-adaptive feature modulation. Following [4], the SDXL inpainting model [1] is introduced as our base diffusion model, and the UNet of SDXL [28] as the garment net.

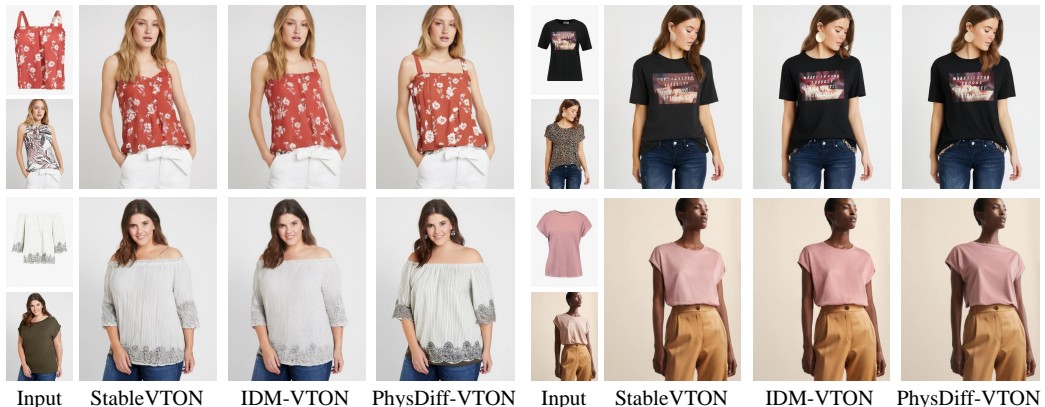

| Input | StableVTON | IDM-VTON | PhysDiff-VTON | Input | StableVTON | IDM-VTON | PhysDiff-VTON |

Figure 1: Qualitative comparison. Our method can generate more natural and physically accurate distortions and wrinkles.

Table 2: Hyper-parameter analysis. We evaluate our method on the VITON-HD dataset under different numbers of pose-aware deformation features, wavelet transformation injection position, potential regularization strength $\lambda$, and maximum noise level $\sigma_{max}$. For simplicity, we only present LPIPS and FID metrics.

<table>
<tr><td colspan="3">(a) Feature Numbers.</td><td colspan="3">(b) Wavelet inject.</td><td colspan="3">(c) Reg. Strength.</td><td colspan="3">(d) Noise level.</td></tr>
<tr><td>nums</td><td>LPIPS</td><td>FID</td><td>pos</td><td>LPIPS</td><td>FID</td><td>$\lambda$</td><td>LPIPS</td><td>FID</td><td>$\sigma_{max}$</td><td>LPIPS</td><td>FID</td></tr>
<tr><td>5</td><td>0.101</td><td>6.30</td><td>$\mathrm{pre}_{res}$</td><td>0.103</td><td>6.32</td><td>5</td><td>0.099</td><td>6.29</td><td>0.8</td><td>0.100</td><td>6.31</td></tr>
<tr><td>10</td><td>0.093</td><td>6.21</td><td>$\mathrm{pre}_{att}$</td><td>0.096</td><td>6.24</td><td>10</td><td>0.093</td><td>6.21</td><td>1.0</td><td>0.096</td><td>6.25</td></tr>
<tr><td>15</td><td>0.092</td><td>6.24</td><td>$\mathrm{pre}_{out}$</td><td>0.093</td><td>6.21</td><td>15</td><td>0.104</td><td>6.32</td><td>1.2</td><td>0.093</td><td>6.21</td></tr>
<tr><td>20</td><td>0.092</td><td>6.22</td><td></td><td></td><td></td><td>20</td><td>0.109</td><td>6.38</td><td>1.4</td><td>0.104</td><td>6.36</td></tr>
</table>

## 4 Experiments

### 4.1 Experimental Setup

**Datasets** We conduct comprehensive evaluations on VITON-HD [3] and DressCode [24]. We train our model on the VITON-HD dataset, which contains 11,647 person-garment image pairs.

**Metrics** Our quantitative analysis employs four complementary measures, i.e., *LPIPS* [47] for perceptual similarity, *SSIM* [39] assessing structural preservation, *FID* [11] evaluating distributional alignment, and *CLIP-I* [29] quantifying semantic consistency.

**Baselines** We compare against two architectural paradigms following [4]. HR-VITON [19] and GP-VTON [40] representing GAN-based approaches. Diffusion-based LaDI-VTON [23], DCI-VTON [8], StableVITON [18] and IDM-VITON [4] utilizing latent garment conditioning. All baselines are evaluated at native 1024×768 resolution using official implementations.

### 4.2 Results and Analysis

**Qualitative Evaluation** Fig. 1 illustrates the qualitative comparison of our method with StableVI-TON [18] and IDM-VITON [4]. Our physically aware warping technology maintains physically correct distortion where fabrics are stacked, even when the baseline wrinkles unnaturally. The wavelet enhancement process successfully preserves details such as textures, patterns, and text that are difficult to capture with other methods. PRPO trajectory optimization prevents unnatural stretching in silk materials that diffusion baselines struggle with.

**Quantitative Comparison** As Tab. 1 shows, PhysDiff-VTON achieves state-of-the-art performance across all metrics. Particularly noteworthy is the LPIPS improvement over IDM-VITON on VITON-HD, demonstrating the effectiveness of our frequency-aware architecture for full-body outfits. The FID reduction on VITON-HD confirms enhanced physical plausibility through continuum mechanics

Table 3: Ablation study. Contribution of each component evaluated by removing it in terms of LPIPS, SSIM, FID, and CLIP-I.

| Dataset | VITON-HD | | | | DressCode | | | |
|---|---|---|---|---|---|---|---|---|
| Method | LPIPS↓ | SSIM↑ | FID↓ | CLIP-I↑ | LPIPS↓ | SSIM↑ | FID↓ | CLIP-I↑ |
| *w/o Deform* | 0.102 | 0.873 | 6.33 | 0.882 | 0.069 | 0.922 | 8.51 | 0.898 |
| *w/o Wavelet* | 0.098 | 0.868 | 6.24 | 0.888 | 0.062 | 0.916 | 8.40 | 0.904 |
| *w/o PRPO* | 0.096 | 0.875 | 6.25 | 0.890 | 0.059 | 0.925 | 8.33 | 0.912 |
| *PhysDiff-VITON* | **0.093** | **0.881** | **6.21** | **0.894** | **0.055** | **0.932** | **8.27** | **0.918** |

Table 4: Efficiency of our method. PCMA represents peak CUDA memory allocated, and TOI represents the time of a batch inference (resolution: $1024 \times 768$, steps=30, batch size=2).

| | StableVITON | IDM-VITON | PhysDiff-VITON |
|---|---|---|---|
| PCMA(G) | 13.105 | 25.183 | 25.245 |
| TOI(s) | 29s | 11s | 11s |

modeling. CLIP-I gains highlight the superior semantic alignment between the generated garments and the target garments.

### 4.3 Hyperparameter Analysis

We conduct systematic hyperparameter studies on the VITON-HD validation set to evaluate four critical design choices. All experiments use the same training protocol with a batch size 32 and 200K iterations. We evaluate our method on the VITON-HD dataset under different number of pose-aware deformation features, wavelet transformation injection position, potential regularization strength $\lambda$, and maximum noise level $\sigma_{max}$. We empirically select the values of these hyperparameters based on the experimental results in Tab. 2.

### 4.4 Ablation Studies and Efficiency

To validate our core innovations, we systematically disable individual components in Tab. 3. For VITON-HD, removing pose-guided deformation (*w/o Deform*) causes severe FID degradation, as rigid warping fails to simulate fabric dynamics. Disabling wavelet decomposition (*w/o Wavelet*) increases SSIM, demonstrating its critical role in preserving high-frequency textures. The absence of potential-aware sampling (*w/o PRPO*) increases LPIPS due to temporal inconsistency in denoising trajectories. Full implementation achieves optimal balance across all metrics. We also investigate the algorithm efficiency regarding batch inference time and CUDA memory consumption, as shown in Tab. 4.

## 5 Conclusion

We introduce PhysDiff-VTON, a physics-integrated diffusion framework that harmonizes fabric dynamics and spectral fidelity in image-based virtual try-on. The proposed method bridges continuum mechanics with generative modeling through pose-guided deformable warping, which simulates strain-dependent garment deformations while preserving local curvature continuity. Complementing this, wavelet-constrained decomposition explicitly safeguards high-frequency textile patterns via frequency-adaptive attention, overcoming spectral erosion inherent in iterative denoising. A novel trajectory optimization strategy further enhances spatiotemporal coherence by reformulating diffusion sampling as an energy-minimized variational process, ensuring anatomically consistent drapery evolution. Comprehensive experiments validate the framework's superiority in synthesizing geometrically plausible and texture-faithful results under challenging articulations, establishing new theoretical connections between physical simulation and diffusion-based synthesis. This work advances virtual try-on toward practical deployment while offering a blueprint for physics-aware generative models in dynamic image synthesis tasks.

**Acknowledgement.** This work is supported by the Science and Technology Commission of Shanghai Municipality under research grant No. 25ZR1401187.

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

## A   Theoretical Rationale for PRPO

The theoretical justification for the Potential-Regularized Path Optimization (PRPO) can be comprehensively analyzed through its foundational connections to stochastic optimal control, compatibility with probabilistic evolution equations, and consistency in discrete implementations. At its core, PRPO reinterprets the reverse process of the diffusion model through the lens of path integral control theory, where the action functional

$$\mathcal{S}[x(t)] = \mathbb{E}\left[\|s_\theta(x(t), t) - \nabla_x \log q_t(x(t))\|^2 + \lambda E(x) + \sigma \|\xi(t)\|^2\right]$$

encodes a trade-off between score-matching fidelity, domain-specific regularization, and controlled stochastic exploration. Minimizing this action corresponds to selecting the most probable paths under the Onsager-Machlup formalism [9], where the kinetic matching term ensures adherence to the data manifold, the potential term $E(x)$ imposes soft constraints like smoothness or physical consistency, and the noise energy term modulates exploration sensitivity. The derived modified stochastic differential equation (SDE)

$$dx = \left[f - g^2 s_\theta + \lambda g^2 \nabla_x E\right] dt + \sigma g d\bar{w}$$

maintains theoretical consistency through its Fokker-Planck equation [30]

$$\frac{\partial p_t}{\partial t} = -\nabla \cdot \left(\left[f - g^2 s_\theta + \lambda g^2 \nabla_x E\right] p_t\right) + \frac{\sigma^2 g^2}{2} \nabla^2 p_t,$$

where bounded regularization strength $\lambda$ preserves the contraction properties of the primary drift term, and controlled noise intensity $\sigma$ satisfies Novikov's condition [7] to maintain measure equivalence between forward and reverse processes. Discretization analysis reveals that the PRPO update step

$$x_{t-1} = \frac{1}{\sqrt{\alpha_t}} \left(x_t - \frac{\beta_t}{\sqrt{1 - \bar{\alpha}_t}} \epsilon_\theta(x_t, t)\right) + \lambda \beta_t \nabla_x E(x_t) + \sigma \sqrt{\beta_t} z$$

achieves $O(\beta^{3/2})$ approximation error through Itō-Taylor expansion [16], matching conventional diffusion model discretization accuracy while introducing non-invasive regularization. The potential function $E(x)$ operates under weak interference principles ($\|\lambda \nabla E\| \ll \|s_\theta\|$) and manifold preservation constraints, ensuring adjustments remain proximal to the support of the data distribution. Dynamic noise annealing via $\sigma(t) = \sigma_{\max} e^{-kt}$ implements simulated annealing-inspired exploration, probabilistically converging to global minima while enabling early-stage diversity exploration. Crucially, PRPO's inference-time adaptation paradigm preserves pretrained score networks, unlike energy-based fine-tuning methods, achieving task-specific regularization through deterministic path optimization rather than retraining. This synthesis of variational action minimization with controlled stochastic dynamics provides a rigorous mathematical grounding while maintaining practical flexibility across domains.

## B   From Action Functional to Reverse-time SDE

The derivation of the modified reverse-time dynamics equation from the action functional can be systematically explained through variational optimization within the stochastic calculus framework. Starting with the action functional

$$\mathcal{S}[x(t)] = \int_0^T \|s_\theta(x, t) - \nabla_x \log q_t(x)\|^2 dt + \lambda \int_0^T E(x) dt + \sigma \int_0^T \|\xi(t)\|^2 dt,$$

we parameterize the reverse process dynamics by a stochastic differential equation (SDE) $dx = a(x, t)dt + b(x, t)dw$, where the drift term $a(x, t)$ and diffusion term $b(x, t)$ are optimized to minimize $\mathcal{S}[x(t)]$. The first term in $\mathcal{S}$, enforcing score matching, directly recovers the conventional reverse drift $a(x, t) = f(x, t) - g(t)^2 s_\theta(x, t)$ through the equivalence between score matching loss minimization and drift correction.

Variational analysis of the potential regularization term $\lambda \int E(x)dt$ introduces an additional gradient correction: perturbing the path $x(t) \to x(t) + \delta x(t)$ yields a variation

$$\delta \left( \lambda \int E(x)dt \right) = \lambda \int \nabla_x E(x) \cdot \delta x \, dt,$$

which corresponds to augmenting the drift with $\lambda g(t)^2 \nabla_x E(x)$, scaled by the noise coefficient $g(t)^2$ from the original diffusion process. Simultaneously, the stochastic control term $\sigma \int \|\xi(t)\|^2 dt$ regulates noise energy via optimal control theory, leading to a diffusion term adjustment $b(x,t) = \sigma g(t)$ that preserves the Wiener process structure while modulating exploration intensity.

Combining these contributions, the optimized SDE becomes,

$$dx = \left[ f(x,t) - g(t)^2 s_\theta(x,t) + \lambda g(t)^2 \nabla_x E(x) \right] dt + \sigma g(t) d\bar{w}.$$

The compatibility of this modified dynamics with the target distribution $q_0(x)$ is verified through its Fokker-Planck equation [30],

$$\frac{\partial p_t}{\partial t} = -\nabla_x \cdot \left( \left[ f - g^2 s_\theta + \lambda g^2 \nabla_x E \right] p_t \right) + \frac{\sigma^2 g^2}{2} \nabla_x^2 p_t,$$

where the original reverse process is recovered when $\lambda = 0$ and $\sigma = 1$. Theoretical consistency requires bounded regularization strength $\lambda$ to avoid destabilizing the primary drift term and adherence to Girsanov's theorem [7] for noise intensity $\sigma g(t)$. This derivation rigorously unifies score matching, potential-guided regularization, and controlled stochasticity within a single variational framework, establishing the mathematical foundation for the path optimization mechanism of PRPO.

## C   From Reverse-time SDE to PRPO Sampling

In a variance-preserving forward diffusion governed by the SDE

$$dx = -\tfrac{1}{2}\beta(t) \, x \, dt + \sqrt{\beta(t)} \, dw,$$

we discretize with $\Delta t = 1$ according to the DDPM [12] parametrization $\beta_t = \beta(t)$, $\alpha_t = 1 - \beta_t$, and $\bar{\alpha}_t = \prod_{s=1}^{t} \alpha_s$. The corresponding reverse-time SDE takes the form

$$dx = \left[ -\tfrac{1}{2}\beta(t)x - \beta(t)s_\theta(x,t) \right] dt + \sqrt{\beta(t)} \, d\bar{w},$$

where $s_\theta(x,t) = -\epsilon_\theta(x,t)/\sqrt{1 - \bar{\alpha}_t}$. Introducing an energy-gradient correction and a controllable noise term yields the modified dynamics

$$dx = \underbrace{\left[ -\tfrac{1}{2}\beta(t)x - \beta(t)s_\theta(x,t) \right] dt}_{\text{standard reverse drift}} + \underbrace{\lambda \, \beta(t) \, \nabla_x E(x) \, dt}_{\text{energy correction}} + \underbrace{\sigma \, \sqrt{\beta(t)} \, d\bar{w}}_{\text{controllable noise}}.$$

Applying the Euler–Maruyama scheme [21] with $\Delta t = 1$ to each term gives for the step $t \to t-1$ the updates

$$-\tfrac{1}{2}\beta_t x_t - \beta_t s_\theta(x_t,t) = -\tfrac{\beta_t}{2}x_t + \tfrac{\beta_t}{\sqrt{1-\bar{\alpha}_t}}\epsilon_\theta(x_t,t) \quad \implies \quad \frac{1}{\sqrt{\alpha_t}}\left( x_t - \tfrac{\beta_t}{\sqrt{1-\bar{\alpha}_t}}\epsilon_\theta(x_t,t) \right),$$

$$\lambda \, \beta_t \, \nabla_x E(x_t),$$

and

$$\sigma \, \sqrt{\beta_t} \, z, \quad z \sim \mathcal{N}(0, I).$$

Thus, one arrives at the PRPO sampling rule

$$x_{t-1} = \frac{1}{\sqrt{\alpha_t}}\left( x_t - \frac{\beta_t}{\sqrt{1-\bar{\alpha}_t}}\epsilon_\theta(x_t,t) \right) + \lambda \, \beta_t \, \nabla_x E(x_t) + \sigma \, \sqrt{\beta_t} \, z.$$

Matching coefficients confirms that the discrete energy-gradient term $\lambda\beta_t\nabla_x E$ and noise coefficient $\sigma\sqrt{\beta_t}$ exactly reflect their continuous-time origins, while the discretization error remains $O(\beta_t^{3/2})$, ensuring numerical stability provided $\lambda\beta_t\|\nabla E\|$ remains small relative to the deterministic update and $\sigma\sqrt{\beta_t}$ decays appropriately.

# D Numerical Stability

In order to ensure numerical stability in the PRPO discrete update

$$x_{t-1} \;=\; \frac{1}{\sqrt{\alpha_t}}\left(x_t - \frac{\beta_t}{\sqrt{1-\bar{\alpha}_t}}\epsilon_\theta(x_t,t)\right) \;+\; \lambda\,\beta_t\,\nabla_x E(x_t) \;+\; \sigma(t)\,\sqrt{\beta_t}\,z,$$

we require two conditions. First, the energy-correction term must remain bounded, which entails

$$\lambda\,\beta_t\,\|\nabla_x E(x_t)\| \;\ll\; \left\|\frac{\beta_t}{\sqrt{\alpha_t(1-\bar{\alpha}_t)}}\,\epsilon_\theta(x_t,t)\right\|.$$

Dividing both sides by $\beta_t$ (with $\beta_t > 0$) gives

$$\lambda\,\|\nabla_x E(x_t)\| \;\ll\; \frac{\|\epsilon_\theta(x_t,t)\|}{\sqrt{\alpha_t(1-\bar{\alpha}_t)}}.$$

Under the common score-matching assumption $\|\epsilon_\theta(x_t,t)\| \propto \sqrt{1-\bar{\alpha}_t}$, this further simplifies to

$$\lambda\,\|\nabla_x E(x_t)\| \;\ll\; \frac{1}{\sqrt{\alpha_t}},$$

so that when $\alpha_t \to 0$ one must choose $\lambda$ sufficiently small or design $E(x)$ so that $\|\nabla_x E(x_t)\|$ decays naturally.

Second, the noise amplitude must be controllable by designing $\sigma(t)$ to decay over $t$. If we set

$$\sigma(t) = \sigma_{\max}e^{-kt},$$

then the variance of the noise term $\sigma(t)\sqrt{\beta_t}\,z$ is

$$\mathrm{Var}\big(\sigma(t)\sqrt{\beta_t}\,z\big) = \sigma_{\max}^2 e^{-2kt}\,\beta_t\,I,$$

and requiring $\lim_{t\to 0}\sigma_{\max}^2 e^{-2kt}\beta_t = 0$ guarantees that as $t \to 0$ (late in generation) the stochastic perturbation vanishes. For example, if $\beta_t = (\beta_{\max}/T)\,t$ (a linear schedule), then one enforces

$$e^{-2kt}\,\frac{\beta_{\max}\,t}{T} \le \varepsilon$$

by tuning $k$ and $\sigma_{\max}$.

Together, these two requirements,

$$\lambda\,\|\nabla_x E(x_t)\| \ll \frac{1}{\sqrt{\alpha_t}} \quad \text{and} \quad \sigma(t) = \sigma_{\max}e^{-kt}$$

, ensure that the PRPO algorithm remains numerically stable by balancing the deterministic score-matching update against energy-based correction and diminishing noise.

