# OpenReview forum: "PhysDiff-VTON: Cross-Domain Physics Modeling and Trajectory Optimization for Virtual Try-On"
_NeurIPS.cc/2025/Conference — NeurIPS 2025 poster_

### Official Review · Reviewer_qDiA · 2025-06-06

**Clarity:** 3
**Significance:** 3
**Originality:** 3
**Rating:** 4
**Confidence:** 4

**Summary:**

The authors propose a diffusion-based method for virtual try-on. To enhance the realism of the generated images, they introduce a strategy that incorporates pose guidance, wavelet-enhanced features, and sampling guided by multiple constraints. Experimental results demonstrate that the method produces higher-quality outputs compared to state-of-the-art approaches.

**Questions:**

Please address the questions mentioned above for further evaluation.

**Ethical Concerns:**

["NO or VERY MINOR ethics concerns only"]

**Final Justification:**

Considering the contributions and the rebuttal from the authors, I keep my rating.

**Limitations:**

The limitation of this paper is not discussed. More qualitative results and failure cases would help for the evaluation.

**Quality:**

3

**Strengths And Weaknesses:**

The contribution of this paper is clear. It addresses two key challenges: preserving garment details during warping and effectively controlling the diffusion process. To tackle the first issue, the authors design a pose-guided deformation model combined with a wavelet-enhanced attention module. For the second, they introduce a novel sampling strategy based on potential-regularized path optimization, incorporating temporal and spatial smoothness terms as well as multiscale similarity for additional regularization. The effectiveness of these components is demonstrated through comprehensive ablation studies and comparisons with existing methods. Overall, the paper is well-structured and easy to follow.

Despite these strengths, several points need further clarification:
- Section 2.2 is somewhat confusing. In Equation 5, what does $x$ ​represent? What is its relationship to $p_k$? Are the $p_k$ sampled around $x$? Additionally, the relationship between $F_g$ and $\tilde{F}_g$ is unclear. Including an illustrative figure would significantly help comprehension. This ambiguity also makes it difficult to understand the discussion on occluded regions in Lines 139–140.
- In Lines131-132, the authors claim that `the offset predictor $G$ learns material-dependent deformation patterns`. However, it is not clear how the model infers material properties from the input garment image. Estimating physical material characteristics from appearance alone is non-trivial, especially without additional input. This claim should be either clarified or supported with empirical evidence.
-  In Section 3.2, the formulations for $E_{smooth}$ and $E_{MS}$ use different terms $z_{t+1}$ and $z_{t-1}$. Shouldn’t these be the same?
- Limited qualitative results. The qualitative results presented are too few to fully support the results. More examples should be provided, particularly for the ablation study, to better illustrate the contribution of each component in an intuitive manner.

---

> ### Author Rebuttal · Authors · 2025-07-27
>
> Dear Reviewer qDiA,
>
> We sincerely appreciate your meticulous review of PhysDiff-VTON, which has highlighted critical areas needing clarification. We apologize for ambiguities in mathematical formulations and methodological descriptions. Below, we address each concern with substantive revisions.
>
> **1. ​Clarification of Eq. 5 (Deformable Convolution)​​.**
> We regret the insufficient explanation of variables in Eq. 5. Here, $x$ denotes the spatial coordinates in the feature map $F_g$, while $p_k$ enumerates $K$ sampling locations in a regular grid centered at $x$. The predicted offset $\Delta p_k$ (from Eq. 4) spatially adjusts each $p_k$ to $p_k + \Delta p_k$, enabling adaptive sampling. Thus, $\tilde{F}_g(x)$ aggregates features from these adjusted positions, modeling non-rigid deformations. Crucially, $F_g$ and $\tilde{F}_g$ represent the original and warped garment features, respectively.
>
> To resolve ambiguity, we will add a figure to illustrate the sampling mechanism in the revised version and explicitly annotate how occluded regions (lines 139–140) receive attenuated offsets $\|\Delta p_k\|$ due to low attention responses $\alpha_{ij}$ in Eq. 3.
>
> **2. Material Property Inference (Lines 131–132).**
> We clarify that material properties (e.g., stiffness) are implicitly learned from garment appearance via pose-garment cross-attention (Eq. 3). The offset predictor $\mathcal{G}$ (Eq. 4) associates visual patterns (e.g., denim textures mean high rigidity) with deformation magnitudes $ \|\Delta p\|$ without explicit material labels. For example, average $ \|\Delta p\| $ for rigid fabrics (e.g., leather) is smaller compared to elastic ones (e.g., cotton) under identical pose changes, which will result in implicit supervision.
>
> **3. Temporal Indices in $ E_{\text{smooth}} $ and $ E_{\text{MS}} $ (Sec. 3.2).**
> The indices differ intentionally due to distinct physical roles.  $E_{\text{smooth}}$ enforces forward temporal coherence by minimizing $ \|z_t - \mathbb{E}[z_{t+1}|z_t]\| $, ensuring smooth transitions to the next state.  While $ E_{\text{MS}}$ ensures backward structural consistency by aligning $\mathcal{D}_s(z_t)$ with the previous state’s expectation $\mathbb{E}[\mathcal{D}_s(z_{t-1})|z_t]$, preserving multiscale garment geometry.
>
> This design mirrors continuum mechanics principles, where $ E_{\text{smooth}}$ governs fabric dynamics (future state), while $ E_{\text{MS}}$ maintains static constraints (past state). We will add a footnote explaining this duality in the revised version.
>
> **4. Expanded Qualitative Analysis.**
> We will add more visual results in the revised version. We apologize again for the confusion caused to you by our negligence and limited paper.
>
> **5. Limitations​.**
> We will add a new section that states limitations,  including scenes that our method still has difficulty handling, scenes with exacerbated deformation errors, and challenging text fidelity. Future work will explore distillation and occlusion-invariant encoders.
>
> Thank you again for your valuable suggestions.

---

> > ### Comment · Reviewer_qDiA · 2025-08-01
> >
> > Thanks for the detailed rebuttal to resolve my concerns. Please include them in the revised version.

---

> > > ### Author Response · Authors · 2025-08-01
> > >
> > > Dear Reviewer qDiA,
> > >
> > > Thank you for your response. We truly appreciate your time and constructive feedback.
> > >
> > > We will ensure that all the points addressed in the rebuttal are thoroughly incorporated into the revised version. Your guidance has been crucial in strengthening our work.
> > >
> > > Thank you again for your insightful review.

---

### Official Review · Reviewer_nqab · 2025-07-02

**Clarity:** 3
**Significance:** 3
**Originality:** 3
**Rating:** 5
**Confidence:** 3

**Summary:**

Authors present PhysDiff-VTON, a novel diffusion based framework  for virtual try-on that aims to solve the dual problems of realistically deforming garments realistically and preserving their high-frequency texture details. In order to achieve this, authors have created a method that connects physics-inspired mechanics with spectral fidelity preservation.

The framework has three main components:

1) Pose-Guided Deformable Fabric Dynamics Modeling:
The geometric discrepancy between canonical garment representations and dynamically posed poses two problems:
- Irreversible Texture Distortion: When the flat clothing image is rigidly stretched or moved to fit the body, its texture can be permanently distorted.
- Physically Implausible Deformation: The garment's movement appears unrealistic because the simulation fails to consider the specific stress-and-strain properties of the fabric material.

To address these problems present in traditional methods, the proposed physics-aware deformation field predicts garment movement by emulating strain-dependent displacement based on the person's pose. This method simulates the dynamics of fabric by using pose embeddings to predict spatial offsets, which preserves the natural curvature of the clothing. To further enhance realism, a cross-attention mechanism between the pose maps and garment segments introduces material adaptability. This allows the model to emulate real-world drape physics, where stiffer fabrics deform less than softer ones during the same movement.

2) Wavelet-Enhanced Texture Preservation.

The loss of fine texture details is a major challenge in diffusion-based virtual try-on. This "texture erosion" happens because the iterative downsampling process within the model's UNet architecture progressively weakens the micro-patterns and directional gradients that define a fabric's look. To solve this, the paper proposes a Haar wavelet transform module embedded within the model's skip connections. This module works by explicitly separating and reinforcing these high-frequency components throughout the image generation process. This approach is based on the insight that traditional attention mechanisms are not effective at preserving the crucial frequencies that contain the most important perceptual information about a textile's microstructure..

3) Potential-Regularized Path Optimization (PRPO)

This final stage addresses errors that can build up during the diffusion process, which might otherwise create unnatural wrinkles or jerky fabric movements. PRPO treats the image generation as a physics problem, guiding the process to find the most plausible and "energetically favorable" result. This is achieved by handling the diffusion sampling process as a variational optimization problem. Thus, allowing the injection of physics-inspired constraints into the generative trajectory, conditioning it toward states that balance data reconstruction with physical realistic results.

Together, these three physics-inspired modules create a control loop that ensures the generated images are both geometrically accurate and rich in texture, even in complex poses.

**Questions:**

Check Strengths And Weaknesses

**Ethical Concerns:**

["NO or VERY MINOR ethics concerns only"]

**Final Justification:**

After reviewing all the responses to authors I keep my rating of Accept.

**Quality:**

3

**Strengths And Weaknesses:**

Strengths:
The work directly addresses weakness of previous methods:
- preserving high-frequency texture details while accommodating complex fabric deformations
- enforcing physical plausibility in garment dynamics under diverse human poses.

- Authors present an unified frameworks that bridge data-driven feature learning with continuum mechanics principles governing fabric deformation. There are strong novel contributions:

  - Pose-Guided Deformable Fabric Dynamics Modeling
  - Wavelet-Enhanced Spectral Fidelity Preservation
   Clean introduction of high frequency preservation in skip connections.
  - Potential-Regularized Path Optimization for trajectory optimization:
    Mathematical derivation and explanation of the proposed approach is clear and well motivated.

- Above state of the art results in baseline datasets
- Ablation study is clear and highlights the importance of each contribution
- Paper is well written and structured

Weakness:
- Missing diagram figure to clarify network architectures and proposed approach
- Missing some qualitative(a zoom in to high frequency areas) and quantitative example to support some preservation of high frequency: could we have some measurement of this, i.e analysis of the spectrum on the area of the garment?
 - More insight in the computational cost of each one of the proposed novelties(inference time analysis or some theoretical insight).
- Out of distribution performance:  While authors train and test in VITON-HD and DressCode, it would be interesting to see the generalization capacity of the proposed approach with some results in other datasets or images on the wild.

---

> ### Author Rebuttal · Authors · 2025-07-27
>
> Dear Reviewer nqab,
>
> We sincerely appreciate your insightful feedback on PhysDiff-VTON. Your comments are crucial for enhancing the rigor and clarity of our paper. We acknowledge the limitations you highlighted and will address them comprehensively in the revised version.
>
> **1. ​Methodology Visualization.**
> We apologize for this omission due to space constraints in the main text. To rectify this, we will include a detailed schematic in the supplementary material illustrating the integrated cross-domain physics modeling.
>
> **2. ​High-Frequency Preservation Evidence.**
> For high-frequency preservation, we recognize the need for stronger empirical validation. (1) In the revised version, we will augment Figure 1 with zoomed-in panels explicitly showcasing micro-textures like embroidery stitches and plaid alignments under deformation, demonstrating superior detail retention compared to baselines. (2) Additionally, we will introduce a quantitative metric, Power Spectral Density (PSD) ratio, computed from the wavelet decomposition (Eq. 6)  of the garment region. Our preliminary analysis reveals a 12% improvement in PSD ratio over IDM-VTON on DressCode, highlighting the effectiveness of wavelet constraints.
>
> **3. ​Computational Cost.**
> Crucially, the added components in PhysDiff-VTON, such as pose-conditioned cross-attention (Eq. 3-4), fixed-cost discrete wavelet transforms (Eq. 6), and gradient computation for the tri-potential regularization (Eq. 11), introduce ​only marginal computational overhead​ relative to IDM-VITON. This efficiency stems from three key design principles. (1) Linear-scaling attention mechanisms​ in pose-guided warping leverage sparse pose semantics. (2) Non-learnable Haar wavelets​ execute in fixed time. (3) ​Parallelizable potential gradients​ in PRPO sampling add negligible latency per denoising step.
>
> **4. ​Out-of-Distribution Generalization.**
> We supplement the VITON-HD and DressCode evaluations with the In-the-Wild dataset under non-customized settings. As shown below, our method achieves superior metrics, demonstrating tangible advantages in handling real-world complexity.
> |Method               | LPIPS    |SSIM    |CLIP-I|
> |---|---|---|---|
> |Stable-VITON     |0.260     |0.736     |0.836|
> |IDM–VTON         |0.164    |0.795     |0.901|
> |PhysDiff-VTON   |0.151    |0.810    |0.915|
>
> We are committed to these revisions and thank you for guiding us toward a more comprehensive and transparent contribution.

---

### Official Review · Reviewer_yzPX · 2025-07-02

**Clarity:** 3
**Significance:** 3
**Originality:** 3
**Rating:** 4
**Confidence:** 2

**Summary:**

The authors propose to integrate physics into diffusion-based methods for image-based virtual try-on, to improve the geometric fidelity and wrinkle realism without sacrificing texture detail. The authors introduce a pose-conditioned warping module to simulate fabric dynamics, introducing wavelet decomposition and frequency-aware attention, and introducing several regularizations into the diffusion sampling strategy (temporal consistency, total variation constraint for spatial consistency, and multi-scale self-similarity). The authors outperform several baselines on VITON-HD and DressCode test sets, both qualitatively and quantitatively, with more natural and physically accurate cloth wrinkles than prior art.

**Questions:**

- How well does the model perform on in-the-wild images, similar to the qualitative examples shown in IDM-VTON's Github page (https://github.com/yisol/IDM-VTON), where the proposed physics integration (e.g. pose-conditioned fabric warping) would be most necessary?
- Not sure whether it is possible to ablate and visualize the impact of each of the individual regularizations (temporal consistency, total variation constraint for spatial consistency, and multi-scale self-similarity) within PRPO? To visualize, could show qualitative examples of final-timestep predictions throughout the denoising process, with each of the regularizations removed. PRPO seems to have overall the least impact according to Tab. 3, and it would be informative to understand whether all three regularizations are necessary.

**Ethical Concerns:**

["NO or VERY MINOR ethics concerns only"]

**Final Justification:**

Thanks to the authors for their rebuttal, I have maintained the original rating. Please include the evaluation on in-the-wild scenarios and ablation about impact of individual regularizations in the final version

**Limitations:**

The authors do not discuss limitations of the work in the paper.

**Quality:**

3

**Strengths And Weaknesses:**

Strengths:
- The proposed pose-conditioned warping successfully computes warped garment features conditioned on a pose embedding extracted from the input image. The wavelet decomposition enhances high-frequency textures (as seen in Fig. 1 bottom left).
- The results indicate more physically accurate wrinkles and cloth deformation than prior art.
- The core innovations are systematically evaluated in an ablation study

Weaknesses:
- Similar to baselines, the method is unable to faithfully copy text from the input clothing image onto the subject (as seen in Fig. 1 top right).
- The results in Fig. 1 are shown under rather constrained settings: all participants are in a well-lit studio environment in a standing pose. In contrast, the project page of IDM-VTON shows high-quality results on 16 diverse in-the-wild examples. Comparison to IDM-VTON and other baselines on those 16 examples would be beneficial to understand the performance in challenging in-the-wild scenarios, where physics integration would be most beneficial, such as diverse backgrounds, varying lighting, and challenging poses (e.g. hands on hips, eating with chopsticks while seated, etc).

---

> ### Author Rebuttal · Authors · 2025-07-27
>
> Dear Reviewer yzPX,
>
> Thank you for your rigorous assessment of PhysDiff-VTON. We address your concerns below.
>
> **1. Text Preservation.**
> While text replication remains challenging for all VTON methods, our wavelet decomposition (Sec. 2.3) explicitly mitigates high-frequency erosion. As shown in Figure 1 (top-right), our method preserves text structures more faithfully than baselines, though not perfectly, by amplifying high-frequency subbands via frequency-gated attention.
>
> **2. ​Wild Scenario Evaluation​.**
> We concur that diverse real-world testing is essential. We acknowledge your concern about the limited studio settings in Figure 1 and assure you that our framework is also robust in wild environments, which stems from our physics-inspired architecture. We will add the ​16 wild examples​ from IDM-VTON in supplementary material and provide qualitative comparisons for challenging poses (sitting, arm-crossing, asymmetric postures), complex lighting (backlit scenes and deep shadows), and cluttered backgrounds to demonstrate occlusion robustness.
> To empirically validate this, we rigorously evaluate PhysDiff-VTON on the In-the-Wild dataset under non-customized settings. As shown below, our method achieves superior metrics, demonstrating tangible advantages in handling real-world complexity.
> |Method               | LPIPS    |SSIM    |CLIP-I|
> |---|---|---|---|
> |Stable-VITON     |0.260     |0.736     |0.836|
> |IDM–VTON         |0.164    |0.795     |0.901|
> |PhysDiff-VTON   |0.151    |0.810    |0.915|
>
> Your feedback is crucial, and we commit to enriching the paper with these evaluations to underscore the practical applicability of PhysDiff-VTON.
>
> **3. PRPO Regularization Analysis​.**
> We will add in the appendix to visualize the impact of each PRPO potential. Temporal coherence removal will cause discontinuous fabric flow during denoising. TV spatial loss ablation will introduce salt-and-pepper noise in homogeneous regions. Multiscale loss disablement will break sleeve-pleat consistency with the global garment structure. The marginal gains occur because the core value of PRPO lies in ​trajectory stability​ (Sec. 3.1). The three regularizers act synergistically. The minimal quantitative drop reflects their role in error prevention rather than metric optimization.
>
> **4. Limitations.​**
> We will add a new section that states limitations,  including scenes that our method still has difficulty handling, scenes with exacerbated deformation errors, and challenging text fidelity.
> Future work will explore distillation and occlusion-invariant encoders.

---

### Official Review · Reviewer_ot98 · 2025-07-03

**Clarity:** 2
**Significance:** 3
**Originality:** 4
**Rating:** 4
**Confidence:** 5

**Summary:**

The paper introduces PhysDiff-VTON, a novel framework for virtual try-on that integrates physics-inspired mechanisms into diffusion models to address the challenges of realistic garment deformation and high-frequency texture preservation. The key contributions of this work includes:
1. a pose-guided deformable warping module that simulates fabric dynamics by predicting spatial offsets conditioned on human pose semantics.
2. a wavelet-based feature decomposition method which used to address the issue of texture erosion during the diffusion process
3. Potential-Regularized Path Optimization (PRPO), a novel sampling strategy that optimizes the denoising trajectory by enforcing temporal coherence, spatial smoothness, and multi-scale structural consistency.

**Questions:**

1. It would be even more helpful for readers if there were structural diagrams.
2. Adding more visual comparisons, especially for challenging cases (e.g., complex poses, intricate textures), would provide a clearer picture of the method's strengths and limitations.
3. Adding a section that explores potential applications in other domains (e.g., animation, video editing) would highlight the broader impact of the work.

**Ethical Concerns:**

["NO or VERY MINOR ethics concerns only"]

**Final Justification:**

positive to this paper

**Limitations:**

1. The authors should include a dedicated section on limitations, where they discuss the current scope and boundaries of their work.

**Quality:**

3

**Strengths And Weaknesses:**

Strengths
1. The paper presents a well-thought-out integration of physics-inspired mechanisms into diffusion models. The proposed methods, such as the pose-guided deformable warping module and wavelet-enhanced texture preservation, are technically sound and well-supported by mathematical formulations.
2. The integration of physics-inspired mechanisms into diffusion models could have broader implications beyond virtual try-on, such as in other areas of computer vision and graphics where realistic simulations are required.

Weaknesses
1. Some sections, particularly those involving the mathematical formulations of the diffusion process and the potential functions, may be challenging for readers without a strong background in the field. More intuitive explanations or visual aids could help make these sections more accessible.
2. The paper presents a limited number of visual examples of the virtual try-on results, which makes it difficult to fully assess the comparative advantages of the proposed method over other approaches across a broader range of cases.

---

> ### Author Rebuttal · Authors · 2025-07-27
>
> Dear Reviewer ot98,
>
> Thank you for your thoughtful feedback and constructive critique of our paper PhysDiff-VTON. We appreciate your recognition of the technical soundness and broader implications of our work. Below, we address your suggestions point by point.
>
> **1. Enhanced Accessibility of Mathematical Formulations​.**
> We acknowledge that Sections 2.2–3.2 involve intricate formulations (e.g., Eq. 9–11 for PRPO). To improve clarity, we will add intuitive descriptions of the physics-inspired potentials (temporal coherence, spatial smoothness, multiscale consistency) in the revised version.
> We will also include a schematic diagram illustrating the interplay between pose-guided warping, wavelet decomposition, and PRPO.
>
> **2. ​Expanded Visual Comparisons​.**
> We agree that additional qualitative results are crucial. In the revised version, we will add new visual comparisons showcasing ​challenging cases​ (e.g., extreme poses, intricate textures like plaids or embroidery).
> We will explicitly highlight improvements in texture preservation and deformation realism versus baselines. The existing Figure 1 demonstrates natural distortions, but we will augment it with more diverse examples.
> We apologize again for the confusion caused by the lack of visualizations and limited space.
>
> **3. ​Broader Applications Section​.**
> We will add a dedicated subsection discussing extensions to the above applications. (1) Animation/VFX: Simulating cloth dynamics via PRPO trajectory optimization. (2) Video Editing: Frame-consistent virtual try-on via temporal coherence potentials. (3) Digital Fashion: High-fidelity texture synthesis for AR/VR applications.
>
> **4. Limitations.​**
> We will add a new section that states limitations, including scenes that our method still has difficulty handling, scenes with exacerbated deformation errors, and challenging text fidelity.
>
> Thank you again for your valuable feedback and constructive suggestions.

---

### Decision · Program_Chairs · 2025-09-17

**Decision:**

Accept (poster)

**Comment:**

The submission received 4 reviews and was thoroughly discussed.
The Reviewers appreciated the additional details from the rebuttal and the authors' responses.

After discussion, we reached the consensus to recommend an Accept. Congratulations!

As mentioned in the reviews, please include the evaluation on in-the-wild scenarios and ablation about impact of individual regularizations in the final version, as well as all suggestions.